# Complex Spirals and Pseudo-Chebyshev Polynomials of Fractional Degree

**Paolo Emilio Ricci**

International Telematic University UniNettuno, Corso Vittorio Emanuele II, 39, 00186 Roma, Italy; paoloemilioricci@gmail.com

**Abstract:** The complex Bernoulli spiral is connected to Grandi curves and Chebyshev polynomials. In this framework, pseudo-Chebyshev polynomials are introduced, and some of their properties are borrowed to form classical trigonometric identities; in particular, a set of orthogonal pseudo-Chebyshev polynomials of half-integer degree is derived.

**Keywords:** Bernoulli spiral; Grandi curves; Chebyshev polynomials; pseudo-Chebyshev polynomials; orthogonality property

---

## 1. Introduction

The purpose of this article is to emphasize some simple connections among mathematical objects apparently of different types as the Bernoulli spirals, the Grandi (rhodonea) curves, and the first and second kind Chebyshev polynomials. Namely, by considering polar coordinates and the complex form of the Bernoulli spiral, a straightforward connection between the real and imaginary part of the Bernoulli spiral with the Grandi curves follows. Even the Chebyshev polynomials come out immediately.

Since the rhodonea curves exist even for a fractional index, it is possible to define an extension of the first and second kind Chebyshev polynomials to the case of rational degree. Actually, the resulting functions are not polynomials, but irrational functions. However, several properties of these functions can be derived from their trigonometric definition, by using standard identities of circular functions. In particular, for the function of half-integer degree, $T_{n+1/2}$ and $U_{n+1/2}$, the orthogonality property still holds, in the interval $(-1, 1)$, with respect to the same weight function of their polynomial counterparts.

The second section of the article is devoted to recalling the most simple examples of spirals, including the Archimedes, Bernoulli, Fermat, and other spirals, which can be derived by using an analogy with Cartesian coordinates. Namely, the above-mentioned spirals, considered in the plane $(\rho, \theta)$, correspond to elementary curves in the plane $(x, y)$, which are, respectively, straight lines, exponential, and power functions. This is, possibly, the motivation for the frequent occurrence of spirals or Grandi curves in natural forms (see, e.g., [1,2]).

## 2. Spirals

The Archimedes spiral [3] (Figure 1) has the polar equation:

$$\rho = a\,\theta, \qquad (a > 0, \quad \theta \in \mathbf{R}). \tag{1}$$

If $\theta > 0$, the spiral turns counter-clockwise, if $\theta < 0$, the spiral turns clockwise.

Bernoulli's (logarithmic) spiral [4] (Figure 1) has the polar equation:

$$\rho = a\,b^\theta, \qquad (a, b \in \mathbf{R}^+),$$

$$\theta = \log_b\left(\frac{\rho}{a}\right).$$

(2)

Varying the parameters *a* and *b*, one gets different types of spirals.

The coefficient *a* changes the size, and the term *b* controls it to be "narrow" and in what direction it wraps itself.

Since *a* and *b* are positive constants, some interesting cases are possible. The most studied logarithmic spiral is called harmonic, as the distance between coils is in the harmonic progression whose ratio is $\phi = \frac{\sqrt{5}-1}{2}$, that is, the "golden ratio" relevant to the unit segment.

The logarithmic spiral was discovered by René Descartes in 1638 and studied by Jakob Bernoulli (1654–1705).

Pierre Varignon (1654–1722) called it an equiangular spiral because:

1. The angle between the tangent at a point and the polar radius passing through that point is constant.
2. The angle of inclination with respect to the concentric circles with the center in the origin is also constant.

It is an example of a fractal. As it is written on J. Bernoulli's tomb: Eadem mutata resurgo, but the spiral represented there is of the Archimedes type.

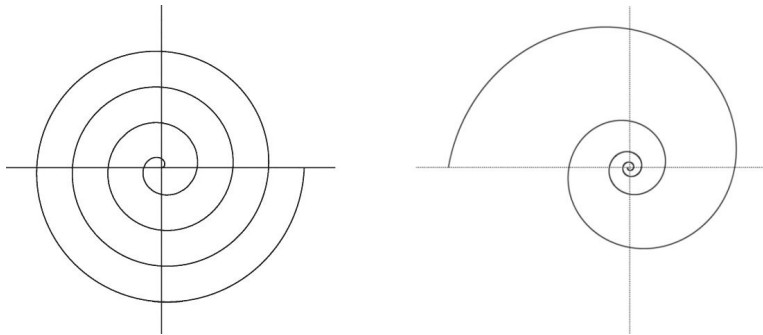

**Figure 1.** Archimedes' vs. Bernoulli's spiral.

Fermat's spiral (or parabolic) (Figure 2) has the polar equation:

$$\rho = \pm a\,\theta^{1/2}.$$

(3)

Fermat's (parabolic) spiral suggests the possibility of introducing intermediate graphs between Archimedes' and Bernoulli's spirals.

In fact, in the plane $(\theta, \rho)$, the graph of Archimedes' spiral is a straight line, while the Bernoulli spiral has an exponential graph, and the Fermat spiral a parabolic graph.

Then, putting:

$$\rho = a\,\theta^{m/n}, \qquad (m, n \text{ positive integers}, \ n \neq 0),$$

(4)

one gets a family of spirals at varying *m* and *n*.

Notice that, if $m > n$, the exponent being greater than one, the coils of the spiral are widening (Figure 2), while if $m < n$, the exponent is less than one, and therefore, the coils of the spiral are shrinking (as in Fermat's case).

Another possibility is to assume $\theta^{m/n}$, where $m/n < 0$ (in this case, the coils are wrapped around the origin (Figure 3) or to use a graph with horizontal asymptotes, in order to get an asymptotic spiral.

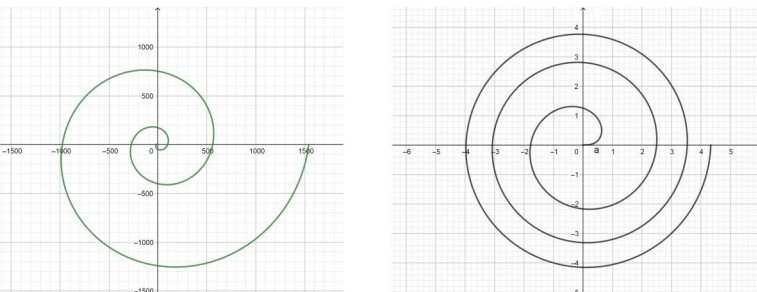

**Figure 2.** Spiral, $\rho = \theta^{3/2}$, and Fermat spiral, $\rho = \theta^{1/2}$.

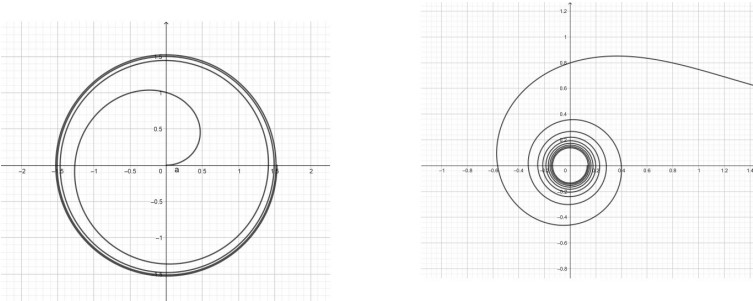

**Figure 3.** Spiral, $\rho = \theta^{-1/2}$, and asymptotic spiral, $\rho = \arctan(\theta)$.

In what follows, we consider a "canonical form" of the Bernoulli spirals assuming $a = 1$, $b = e^n$, that is the simplified polar equation:

$$\rho = e^{n\theta}, \qquad (n \in \mathbf{N}).\tag{5}$$

## 3. The Complex Bernoulli Spiral

We now introduce the complex case, putting:

$$\rho = \Re\rho + i\,\Im\rho,\tag{6}$$

and considering a Bernoulli spiral of the type:

$$\rho = e^{in\theta} = \cos n\theta + i\sin n\theta.\tag{7}$$

Therefore, we have:

$$\rho_1 = \Re\rho = \cos n\theta, \qquad \rho_2 = \Im\rho = \sin n\theta.\tag{8}$$

### 3.1. Rhodonea Curves

The curves defined in polar coordinates by:

$$\rho_1 = \Re\rho = \cos n\theta,\tag{9}$$

are called rhodonea curves or Grandi roses (examples in Figure 4), by the name of Luigi Guido Grandi (1671–1742), who communicated his discovery in a letter to Leibniz in 1713 [5].

Curves of the type:

$$\rho_2 = \Im\rho = \sin n\theta,\tag{10}$$

are essentially equivalent to the preceding ones, up to a rotation of $\frac{\pi}{2n}$ radians.

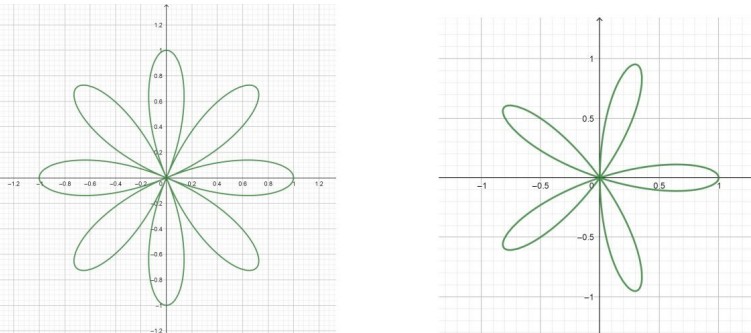

**Figure 4.** Rhodonea, $\rho = \cos(4\theta)$, and rhodonea, $\rho = \cos(5\theta)$.

### 3.2. Chebyshev Polynomials

The Chebyshev polynomials of the first and second kind were introduced by Pafnuty L. Chebyshev (1821–1894). They can be derived as the real and imaginary part of the exponential function $e^{i\,n\,\theta} = (\cos\theta + i\sin\theta)^n$ (see Equation (7)), putting $x = \cos\theta$, and using the Euler formula (see [6] for details).

The first kind Chebyshev polynomials are important in approximation theory and Gaussian quadrature rules. In fact, by using their roots—called Chebyshev nodes—the resulting interpolation polynomial minimizes the Runge phenomenon. Furthermore, the relevant approximation is the best approximation to a continuous function under the maximum norm.

Linked with these polynomials are also the Chebyshev polynomials of the second kind, which appear in computing the powers of $2 \times 2$ non-singular matrices [7]. Generalizations of such polynomials have been also introduced, in particular for computing powers of higher order matrices (see, e.g., [8,9]).

An excellent book is [10]. The importance of these polynomial sets in applications is shown in [11].

Recently, the Chebyshev polynomials of the first and second kind have been used in order to represent the real and imaginary part of complex Appell polynomials [12,13].

The connection of the second kind Chebyshev polynomials with classical polynomials of number theory has been recently underlined by Kim T., Kim D.S. et al. (see, e.g., [14–16]).

Definition of Chebyshev polynomials of the first kind:

$$
\begin{aligned}
T_0(x) &= 1, \\
T_1(x) &= x, \\
T_{n+1}(x) &= 2x\,T_n(x) - T_{n-1}(x),
\end{aligned}
\tag{11}
$$

$$
T_n(x) = \cos(n\,\arccos(x)) \quad \Leftrightarrow \quad T_n(\cos\theta) = \cos(n\,\theta).
$$

Definition of Chebyshev polynomials of the second kind:

$$
\begin{aligned}
U_0(x) &= 1, \\
U_1(x) &= 2x, \\
U_{n+1}(x) &= 2x\,U_n(x) - U_{n-1}(x),
\end{aligned}
\tag{12}
$$

$$
\sqrt{1-x^2}\,U_{n-1}(x) = \sin\left(n\,\arccos(x)\right) \quad \Leftrightarrow \quad U_{n-1}(\cos\theta) = \frac{\sin[n\,\theta]}{\sin\theta}.
$$

As a consequence of the above considerations, it is possible to note the connection of the rhodonea curves with the first kind Chebyshev polynomials.

In fact, the rhodonea curve $\rho = \cos(n\,\theta)$ (Figure 4) can be interpreted as $T_n(\cos\theta)$ (Equation (11)).

In a similar way, the second kind Chebyshev polynomials have as graphical images the roses of the type $U_n(\cos\theta)$ corresponding to $\sin((n-1)\theta)/\sin(\theta)$ (examples in Figure 5).

Note that, in both cases, the rhodonea curve has $n$ petals if $n$ is odd and $2n$ petals if $n$ is even.

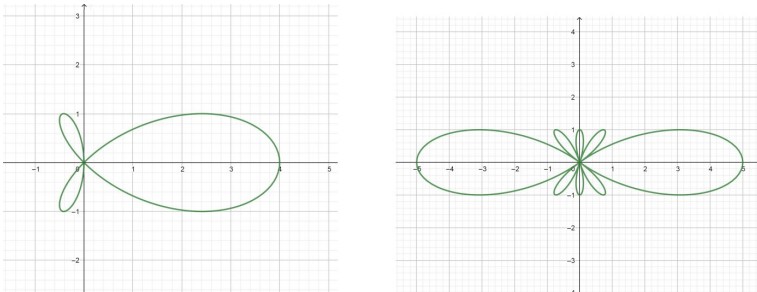

**Figure 5.** $U_4$ *rose*, $\rho = \sin(3\theta)/\sin(\theta)$, and $U_5$ *rose*, $\rho = \sin(4\theta)/\sin(\theta)$.

## 4. Pseudo-Chebyshev Polynomials

The rhodonea curves exist even for rational values of the index $n$ (see, e.g., [17]). This allows us to consider the sets of first and second kind pseudo-Chebyshev polynomials (graphical examples in Figures 6 and 7). The prefix "pseudo" is used because actually, they are not polynomials, but irrational functions, as it is seen in what follows.

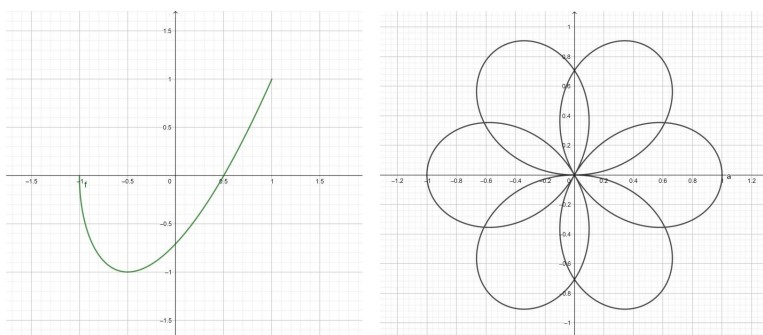

**Figure 6.** Pseudo $T_{3/2} = \cos(1.5\arccos(x))$, and rhodonea, $\rho = \cos(1.5\,\theta)$.

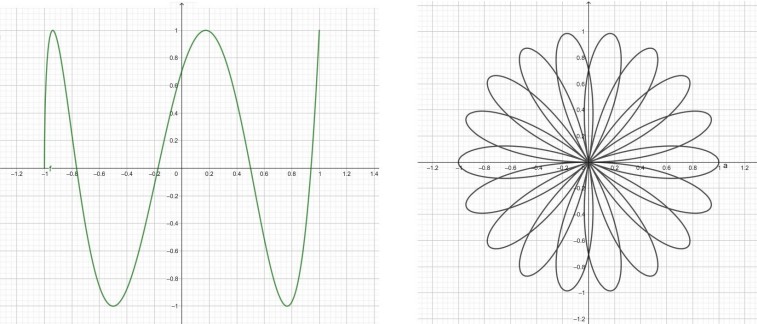

**Figure 7.** Pseudo $T_{9/2} = \cos(4.5\arccos(x))$, and rhodonea, $\rho = \cos(4.5\,\theta)$.

We start assuming the degree in the form $n + \frac{1}{2}$, that is a semi-integer number. This seems to be the most interesting case, since the resulting functions $T_{n+1/2}$ and $U_{n+1/2}$ are proven to be orthogonal, in the interval $(-1, 1)$, with respect to the same, corresponding weights, of the first and second kind Chebyshev polynomials.

We put, by definition:

$$T_{k+\frac{1}{2}}(x) = \cos\left(\left(k + \tfrac{1}{2}\right)\arccos(x)\right) \tag{13}$$

$$\sqrt{1 - x^2}\, U_{k-\frac{1}{2}}(x) = \sin\left((k + \tfrac{1}{2})\, \arccos(x)\right).$$

(14)

**Remark 1.** *It is worth noting that the third and fourth kind Chebyshev polynomials $V_n(x)$ and $W_n(x)$ (see, e.g., [18]) have a similar definition, but they do not coincide with the pseudo-Chebyshev, since actually they are true polynomials, and satisfy orthogonality properties with respect to different weights (see Figures 8 and 9).*

*The third and fourth kind Chebyshev polynomials have been studied and applied by several scholars (see, e.g., [18–20]), because they are useful in quadrature rules, when the singularities occur only at one of the end points ($+1$ or $-1$) (see [10]). Furthermore, recently, they have been applied in numerical analysis for solving high odd-order boundary value problems with homogeneous or nonhomogeneous boundary conditions [19].*

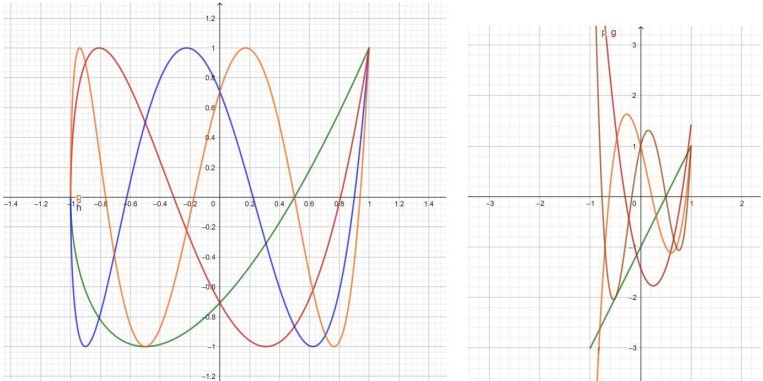

**Figure 8.** Pseudo $T_{n/2}$, $n = 3, 5, 7, 9$, and third kind $V_k(x)$ $k = 1, 2, 3, 4$.

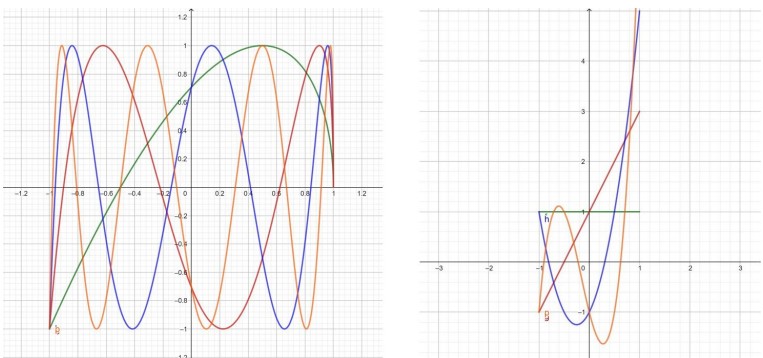

**Figure 9.** Pseudo $U_{n/2}$, $n = 1, 3, 5, 7$, and fourth kind $W_k(x)$ $k = 0, 1, 2, 3$.

*4.1. The Case of Half-Integer Degree*

In particular, we have:

$$T_{1/2}(x) = \cos\left(\tfrac{1}{2}\, \arccos(x)\right) = \sqrt{\frac{1 + x}{2}},$$

(15)

$$\sqrt{1 - x^2}\, U_{-1/2}(x) = \sin\left(\tfrac{1}{2}\, \arccos(x)\right) = \sqrt{\frac{1 - x}{2}}.$$

(16)

Therefore, we find:

$$T_{3/2}(x) = \cos\left(\tfrac{3}{2}\arccos(x)\right) = \cos\left(\arccos(x) + \tfrac{1}{2}\arccos(x)\right)$$

$$= \cos(\arccos(x))\cos\left(\tfrac{1}{2}\arccos(x)\right) - \sin(\arccos(x))\sin\left(\tfrac{1}{2}\arccos(x)\right) \tag{17}$$

$$= x\sqrt{\frac{1+x}{2}} - \sqrt{1-x^2}\sqrt{\frac{1-x}{2}},$$

$$T_{5/2}(x) = \cos\left(\tfrac{5}{2}\arccos(x)\right)\cos\left(2\arccos(x) + \tfrac{1}{2}\arccos(x)\right)$$

$$= \cos(2\arccos(x))\cos\left(\tfrac{1}{2}\arccos(x)\right) - \sin(2\arccos(x))\sin\left(\tfrac{1}{2}\arccos(x)\right) \tag{18}$$

$$= T_2(x)\sqrt{\frac{1+x}{2}} - \sqrt{1-x^2}\,U_1(x)\sqrt{\frac{1-x}{2}}.$$

*4.2. Recurrence Relations*

We have, in general:

$$T_{n+1/2}(x) = \cos\left(n\arccos(x) + \tfrac{1}{2}\arccos(x)\right)$$

$$= T_n(x)\sqrt{\frac{1+x}{2}} - \sqrt{1-x^2}\,U_{n-1}(x)\sqrt{\frac{1-x}{2}},$$

that is:

$$T_{n+1/2}(x) = T_n(x)\,T_{1/2}(x) - \left(1 - x^2\right)\,U_{n-1}(x)\,U_{-1/2}(x). \tag{19}$$

In a similar way, for the second kind, we find:

$$U_{n+1/2}(x) = U_{n-1}(x)\,T_{1/2}(x) + U_{-1/2}(x)\,T_n(x). \tag{20}$$

**Remark 2.** *Note that the number of rose petals of the curves $\rho = \cos(\tfrac{n}{2}\theta)$, $n = 1,3,5,\ldots$ is given by the sequence $\{2, 6, 10, 14, 18, 22, \ldots\}$, which appears in the Encyclopedia of Integer Sequences [21] at A016825: positive integers congruent to 2 $\mod 4 : a(n) = 4n + 2$, for $n \geq 0$.*

*4.3. More General Formulas*

By using cosine addition formulas, putting:

$$\frac{m}{n} = \frac{p}{q} + \frac{r}{s}, \tag{21}$$

we find:

$$T_{m/n}(x) = T_{p/q}(x)\,T_{r/s}(x) - \left(1 - x^2\right)\,U_{(p/q)-1}(x)\,U_{(r/s)-1}(x), \tag{22}$$

and by using the sine addition formulas:

$$U_{m/n}(x) = U_{(p/q)-1}(x)\,T_{r/s}(x) + U_{(r/s)-1}(x)\,T_{p/q}(x). \tag{23}$$

Particular Results

$$T_1(x) = T_{1/3}(x)\, T_{2/3}(x) - (1 - x^2)\, U_{-2/3}(x)\, U_{-1/3}(x)\,. \tag{24}$$

$$T_1(x) = \cos\left[3 \cdot \tfrac{1}{3}\arccos(x)\right] = 4\, T_{1/3}^3(x) - 3\, T_{1/3}(x)\,. \tag{25}$$

$$T_2(x) = \cos\left[3 \cdot \tfrac{2}{3}\arccos(x)\right] = 4\, T_{2/3}^3(x) - 3\, T_{2/3}(x)\,. \tag{26}$$

$$T_{2/3}(x) = \cos\left[2 \cdot \tfrac{1}{3}\arccos(x)\right] = 1 - 2\,\sin^2\left[\tfrac{1}{3}\arccos(x)\right]$$

$$= 1 - 2\,(1 - x^2)\, U_{-2/3}(x)\,. \tag{27}$$

$$U_{-1/3}(x) = \frac{\sin\left[2 \cdot \tfrac{1}{3}\arccos(x)\right]}{\sqrt{1 - x^2}}$$

$$= \frac{2}{\sqrt{1 - x^2}}\,\sin\left[\tfrac{1}{3}\arccos(x)\right]\,\cos\left[\tfrac{1}{3}\arccos(x)\right] = 2\, U_{-2/3}(x)\, T_{1/3}(x)\,. \tag{28}$$

$$U_{-2/3}(x) = \frac{\sin\left[\tfrac{1}{3}\arccos(x)\right]}{\sqrt{1 - x^2}} = \sqrt{\frac{1 - T_{1/3}^2(x)}{1 - x^2}}\,. \tag{29}$$

Combining the above equations, we find:

$$T_1(x) = T_{1/3}(x)\, T_{2/3}(x) - 2\, T_{1/3}(x)\left(1 - T_{1/3}^2(x)\right)$$

$$= T_{1/3}(x)\left(2\, T_{1/3}^2(x) + T_{2/3}(x) - 2\right)\,. \tag{30}$$

*4.4. Orthogonality for Half-Integer Degree*

**Theorem 1.** *The Chebyshev functions $T_{m/2}(x)$ satisfy the orthogonality property:*

$$\int_{-1}^{1} T_{m/2}(x)\, T_{n/2}(x)\,\frac{1}{\sqrt{1 - x^2}}\, dx = 0\,, \qquad (m \neq n)\,, \tag{31}$$

*where m,n are positive odd numbers such that m + n = 2k, k = 2, 3, 4, . . . ,*

$$\int_{-1}^{1} T_{m/2}^2(x)\,\frac{1}{\sqrt{1 - x^2}}\, dx = \frac{\pi}{2}\,. \tag{32}$$

**Proof.** As a consequence of Werner formulas, we have, under the above conditions,

$$\int_{-1}^{1} \cos(m/2\arccos(x))\,\cos(n/2\arccos(x))\,\frac{1}{\sqrt{1 - x^2}}\, dx = (\text{putting } x = \cos(2t)\,)$$

$$= 2\int_{0}^{\pi/2} \cos(mt)\,\cos(nt)\, dt = 0\,, \tag{33}$$

and:

$$\int_{-1}^{1} \cos^2(m/2 \arccos(x)) \frac{1}{\sqrt{1-x^2}} \, dx = 2 \int_{0}^{\pi/2} \cos^2(mt) \, dt = \frac{\pi}{2} \, . \tag{34}$$

□

**Theorem 2.** *The Chebyshev functions* $U_{m/2}(x)$ *satisfy the orthogonality property:*

$$\int_{-1}^{1} U_{m/2}(x) \, U_{n/2}(x) \sqrt{1-x^2} \, dx = 0 \, , \qquad (m \neq n) \, , \tag{35}$$

*where m, n are positive odd numbers such that m + n = 2k, k = 2, 3, 4, . . . ,*

$$\int_{-1}^{1} U_{m/2}^2(x) \sqrt{1-x^2} \, dx = \frac{\pi}{2} \, . \tag{36}$$

**Proof.** We have, under the above conditions,

$$\int_{-1}^{1} \sin(m/2 \arccos(x)) \, \sin(n/2 \arccos(x)) \sqrt{1-x^2} \, dx = (\text{putting } x = \cos(2t))$$

$$= 2 \int_{0}^{\pi/2} \sin(mt) \sin(nt) \, dt = 0 \, , \tag{37}$$

and:

$$\int_{-1}^{1} \sin^2(m/2 \arccos(x)) \sqrt{1-x^2} \, dx = 2 \int_{0}^{\pi/2} \sin^2(mt) \, dt = \frac{\pi}{2} \, . \tag{38}$$

□

## 5. Conclusions

The complex form of the Bernoulli spiral, by using Euler formulas, allows us to emphasize connections with Grandi (rhodonea) curves. The rhodonea with the fractional index can be viewed as an extension of first and second kind Chebyshev polynomials to irrational functions. The properties of these "pseudo-Chebyshev functions" are borrowed from classical trigonometric identities. In particular, in the case of half-integer degree, the corresponding functions satisfy the same orthogonality property of the corresponding Chebyshev polynomials of integer degree.

**Funding:** This research received no external funding.

**Acknowledgments:** Dedicated to Hari M. Srivastava with deep admiration.

**Conflicts of Interest:** The author declares no conflict of interest.

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
