# Peer review of "Complex Spirals and Pseudo-Chebyshev Polynomials of Fractional Degree"

_symmetry, doi:10.3390/sym10120671_

Round 1

Reviewer 1 Report

1.    Some relations are not numbered, making it difficult to establish links between them.

2.    There is no interpretation for Rhodonea curve from Figure 6 versus Rhodonea curve from Figure 7.

3.    Also, the difference between Chebyshev rose for Figure 8, versus from Figure 9, should be explained.

4.    What is the origin of polynomials Chebyshev of half-integer degree (see Section 4.1)?

5.    Some editing "glitches" need to be corrected.

6.    Punctuations are used randomly. Insert comma or full stop after each and every equation accordingly (see, for instance, page 8).

7.    References are few and old. I think, the authors  must strengthen the References section with some articles that use some techniques,  to make the techniques used more plausible, for instance: Weak Solutions in Elasticity of Dipolar Porous Materials, Math Probl Eng, 2008 (2008), 1-8, Art. No. 158908; Weak solutions in Elasticity of dipolar bodies with stretch, Carpathian J Math, Vol. 29(1) (2013), pp. 33-40

Author Response

1.   I added numbering of equations.

2.   I added a comment regarding Rhodonea curves.

3.  I added similar comment regarding the Chebyshev polynomials.

4.    I explained how arise the pseudo-Chebyshev, in connection with roses with rational index.

5.   I checked all the manuscript.

6.    I checked puntuations.

7.   I added new references and comments.

In order to shorten the manuscript, I changed the number of figures, by coupling the 

connected ones.

I made references to applications of Chebyshev polynomials is several fields, but it

was impossible to find the Chebyshev polynomials in your references.

Reviewer 2 Report

Chebyshev polynomial is an important field that is applied in various fields such as mathematics and applied mathematics. The author needs to introduce many of these important applications and properties of Chebyshev polynomials to the reader.   It is recommended that the following papers related to this review paper be included in the reference and then introduced to other applications of Chebyshev polynomials for readers. Focusing on the Spirial may not seem very important.

I suggest that the author include the following papers in the references and give some comments

 related to Chebyshev polynomoals

***************************************************

 Kim, T.; Kim, D. S.; Dolgy, D. V.; Kwon, J. Sums of finite products of Chebyshev polynomials of the third and fourth kinds. Adv. Difference Equ. 2018, Paper No. 283, 17 pp.

Kim, T.; Kim, D. S.; Dolgy, D. V.; Park, J.-W. Sums of finite products of Chebyshev polynomials of the second kind and of Fibonacci polynomials. J. Inequal. Appl. 2018, Paper No. 148, 14 pp.

Kim, T; Kim, D. S.; Kwon,J ; Dolgy, D. V. Expressing Sums of Finite Products of Chebyshev Polynomials of the Second Kind and of Fibonacci Polynomials by Several Orthogonal Polynomials, Mathematics 2018, 6(10), 210

Kim, D. S.; Dolgy, D. V.; Kim, T.; Rim, S.-H. Identities involving Bernoulli and Euler polynomials arising from Chebyshev polynomials. Proc. Jangjeon Math. Soc. 15 (2012), no. 4, 361–370.

***************************************************************

After revising this review paper as follows referee's comments, I recommend to publish this paper

in Symmetry.

Author Response

I added your references, in particular in connection with third and fourth kind Chebyshev polynomials.

I underlined the difference between these polynomials and the pseudo-Chebyshev rational functions.

I also upgraded the Reference section, by adding several books and more recent articles, emphasizing the applications of Chebyshev polynomials.

The historical part, about spirals, remain included, since it is useful to introduce the Rhodonea curves of fractional degee, which are the basis for the pseudo-Chebyshev functions.

Round 2

Reviewer 1 Report

No more comments

Reviewer 2 Report

The paper was well revised . I strongly recommend this paper to publish in Symmetry.